**Data Availability Statement:** All relevant data are within the manuscript and its Supporting information files.

**Funding:** The author(s) received no specific funding for this work.

# Who shares fake news on social media? Evidence from vaccines and infertility claims in sub-Saharan Africa

Kerstin Unfried[1]*, Jan Priebe[2]

1 Health Economics Research Group, Bernhard Nocht Institute for Tropical Medicine (BNITM), Hamburg, Germany, 2 BNITM & Hamburg Center for Health Economics (HCHE), Hamburg, Germany

☉ These authors contributed equally to this work.
* kerstin.unfried@bnitm.de

## Abstract

The widespread dissemination of misinformation on social media is a serious threat to global health. To a large extent, it is still unclear who actually shares health-related misinformation deliberately and accidentally. We conducted a large-scale online survey among 5,307 Facebook users in six sub-Saharan African countries, in which we collected information on sharing of fake news and truth discernment. We estimate the magnitude and determinants of deliberate and accidental sharing of misinformation related to three vaccines (HPV, polio, and COVID-19). In an OLS framework we relate the actual sharing of fake news to several socioeconomic characteristics (age, gender, employment status, education), social media consumption, personality factors and vaccine-related characteristics while controlling for country and vaccine-specific effects. We first show that actual sharing rates of fake news articles are substantially higher than those reported from developed countries and that most of the sharing occurs accidentally. Second, we reveal that the determinants of deliberate vs. accidental sharing differ. While deliberate sharing is related to being older and risk-loving, accidental sharing is associated with being older, male, and high levels of trust in institutions. Lastly, we demonstrate that the determinants of sharing differ by the adopted measure (intentions vs. actual sharing) which underscores the limitations of commonly used intention-based measures to derive insights about actual fake news sharing behaviour.

## 1 Introduction

Fake news is a worldwide concern carrying grave political and social consequences [1, 2]. Regarding health, fake news has been linked to worse mental health outcomes, the misallocation of resources, lower compliance with health authorities & regulations, and increases in vaccine hesitancy [3–8]. The substantial influence of misinformation on health beliefs, behavior, and its negative consequences became strikingly visible during the COVID-19 pandemic. For instance, exposure to misinformation about the protective role of smoking and alcohol consumption against COVID-19 infections has been associated with increased smoking and

**Competing interests:** The authors have declared that no competing interests exist.

alcohol use in current consumers. In contrast, people that believed smoking is a risk factor for a severe COVID-19 infection showed greater intentions to quit smoking [9, 10]. In how far people show misinformation-consistent behavior or followed public health recommendations depends among others on the individual level of trust in institutions such as the government and national health authorities [11, 12].

The spread of fake news has often been attributed to the rise of social media platforms which allow for the rapid dissemination of false or misleading information to a broad audience [13–15]. Bearing in mind the continuing rise in social media usage globally and the regular development of innovative and sophisticated techniques to manipulate digital content, a better understanding of the cognitive and behavioral drivers behind the distribution of fake news on social media appears crucial.

In this paper, we conducted online surveys in six sub-Saharan African countries (Ghana, Kenya, Nigeria, South Africa, Tanzania, Uganda) to analyse the individual-level determinants of spreading fake news. More specifically, we examine (deliberate and accidental) actual sharing behavior of vaccine-related fake news—unsubstantiated infertility claims of vaccines for polio, HPV, and COVID-19—on Facebook. As part of the surveys, participants read one fake news article, were asked about their willingness to share it on Facebook, and were ultimately able to share the article. The surveys were conducted in early 2023 and gathered information from 5,307 participants.

Our analysis rests on established conceptual frameworks that distinguish between the accidental and deliberate sharing of fake news. Accidental sharing refers to the distribution of false information due to (i) a person's inability to discern truth (e.g. lacking knowledge or cognitive skills to distinguish truthful from misleading information) or (ii) a lack of motivation to do so (e.g. inattention or psychological biases such as confirmatory bias or motivated reasoning) [16–18]. In contrast, deliberate sharing refers to the purposeful distribution of misinformation. Existing explanations for deliberate sharing imply that other preferences outweigh any preference for truth telling [19, 20].

Only few studies have investigated what determines accidental vs. deliberate sharing of fake news [21–24]. The evidence coming from these studies is far from conclusive, suggesting that the drivers of deliberate and accidental sharing of fake news is context and study-specific. For instance, some studies find that most sharing of fake news can be attributed to accidental sharing [23], while others stress the role of psychological reward mechanisms [25] or the interest in the information as well as claim veracity [26]. Likewise, while some studies find that older people are more likely to share political fake news [27, 28], others find that younger people might do so [21, 23]. There are also contradictory findings regarding the influence of personality traits and demographic variables such as gender and education [21, 24, 25]. Similar context-specific effects are reported from the related literature on individuals' ability to detect misinformation. While there is consensus on some characteristics such as cognitive ability and literacy, for others the empirical evidence is mixed (e.g. age, gender, and income) [18, 23, 29–32].

A major drawback of the current literature is that almost all studies rely on self-reported measures of sharing behavior (intentions) that are notoriously difficult to interpret with respect to actual behavior due to differences related to self-efficacy, social cognition, risk perceptions, task switching, memory capacity, among others [33]. In fact, we are only aware of one study that distinguishes between accidental and deliberate sharing of fake news and utilises actual sharing information, studying the U.S. political context [25].

In this context our study differs from previous research on the role and determinants of accidental and deliberative sharing of fake news in four aspects. First, the existing literature is limited to developed countries and political fake news topics. In contrast, our study examines the sharing of fake news in six low- and middle-income countries (LMICs) in SSA with respect

to vaccinations. Considering that (i) the majority of the world's population lives in LMICs, (ii) the rise in social media usage being particularly pronounced in LMICs, (iii) social media representing a major source of information in LMICs due to lower levels of trust in national governments, (iv) the particularly high susceptibility to fall for fake news in LMICs [34], and (v) vaccine hesitancy, often originating from the consumption of fake news, being among the top ten global health threats [15], our study investigates a pivotal setting that has been largely disregarded. Second, we complement previous research by integrating a more comprehensive set of variables into our analyses. In addition to standard socio-economic controls and personality traits, we in particular examine the role of institutional trust and risk preferences. Third, we are able to investigate actual sharing behavior with click data. In this regard our study provides direct insights into the implications of measuring sharing intentions vs. actual sharing behavior. Bearing in mind that almost all studies are confined to intentional measures, we shed light on the extend and determinants of the related measurement bias. Fourth, by collecting data from six countries and regarding sharing of misinformation related to three distinct vaccines, our study allows us to compare results across contexts.

## 2 Methods

### 2.1 Data collection

For this study we collected primary data via online surveys in six SSA countries (Ghana, Kenya, Nigeria, South Africa, Tanzania, and Uganda). The surveys were hosted on the Uni-Park platform and implemented during February and March 2023. In the following we refer to the surveys as African Health Surveys (AHS). All respondents had to be adults (older than 17 years) and were recruited via Facebook ads. Participation was incentivised with a lottery. Participants could win phone credit of 5GB.

The AHS is an online survey that collects information on participants' health status, health behavior and knowledge, attitudes towards health topics, media usage and other socio-demographic characteristics. As part of the survey, respondents were exposed to an article about the common rumour that vaccines are causing infertility. Participants received randomly one of the three articles that are shown in S2 Appendix. Participants were asked to read the fake news article and about their willingness to share the article on their Facebook account directly afterwards. Ultimately, participants had the opportunity to actually share the article. We observed this behavior with click data. At a later stage of the survey, respondents evaluated the accuracy of the article content.

### 2.2 Ethics statement

This study was approved by the Ethics Committee of the Medical Association of Hamburg, Germany. All participants provided their consent at the beginning of the survey. We have implemented several measures to limit negative consequences related to the exposure to fake news. First of all, the survey contained a thorough debriefing that explained to respondents that they were exposed to an article containing fake news and provided corrective information. Second, the website on which the articles were hosted included a disclaimer on each side in which we visibly highlight that these articles contain misinformation and were used as part of a scientific experiment. Third, the website was shut-down immediately after the finalization of our experiment. Lastly, several tedious steps were implemented before the sharing action. The surveys were carried out in accordance with relevant guidelines and regulations.

## 2.3 Study sample

Our study sample includes 5,307 respondents. We dropped observations that did not finish the survey or could not pass the attention checks. The majority of our respondents are from Kenya (see S2 Table). The average respondent is 29.2 years old (age ranges from 18 to 75 years), male (about 65%), and possesses tertiary education (about 73%). Further descriptive statistics are reported in S1 Table in the supplementary information. In general, the sample appears to be representative of the social media user population. Bearing in mind selection effects into internet access, device possession (smartphone, tablet, pc), social media usage, and willingness to participate in research projects, the sample is not necessarily representative of the national populations in the respective countries.

## 2.4 Construction of variables

**Detection of misinformation.** Detection of misinformation is measured on the basis of respondents' perception of the truthfulness of the article. Respondents assessed the accuracy of the presented article on a 5-point Likert scale ranging from 0 "not true at all" to 4 "entirely true". Based on this information, we classify respondents with a value of 2 and below to be able to detect the misleading information.

**Measures related to the sharing of fake news.** Intentional sharing behavior is measured using the following 1-item survey question "Would you like to share the article on your Facebook account?". Based on respondents' answer to this question we constructed a binary indicator that takes the value 1 if the respondent wanted to share the article. Likewise, actual sharing behaviour is coded as a binary indicator that takes the value 1 if a person actually shared the article on Facebook. To measure whether a person shared the article we rely on click data. More specifically, we recorded a respondent's click on a Facebook share button that linked the survey with the Facebook environment (a respondent's Facebook account). Furthermore, and based on the variables related to actual sharing behaviour and detection of misinformation, we classify respondents into deliberate and accidental sharing types, whereby 'deliberate' refers to respondents that detected the misinformation and shared it and 'accidental' to those who believed in the accuracy of the content and shared the fake news article.

**Individual characteristics.** In our analysis we consider a broad range of individual characteristics that were collected as part of the surveys. First, socio-demographic variables are: age categories (binary indicator variables for age intervals), gender (0 = male, 1 = female), education categories (binary indicator variables for completed education levels), marital status (binary indicator whether person is married), work status ((self-)employed = 1, 0 otherwise), binary indicators for wealth categories (poor, average, rich relative to others). These variables were assessed with self-reported measures. Second, we construct two variables that capture different dimensions of economic preferences: Risk is measured using the well-established 1-item scale on a respondent's willingness to take risk using the following question "Are you a person who is generally willing to take risks?". The item uses a 11-point Likert response scale ranging from 0 "unwilling to take risks" to 10 "very willing to take risks". The item is among others used in the global preference survey [35] and has been shown to be a good predictor of risk behavior in various settings (e.g. [36–38]). Trust is a composite index (average value) that is based on three questionnaire items related to respondents' trust in health-related information from the (i) government, (ii) science and research, and (iii) traditional media. Responses to each of the three items are recorded on a 4-point Likert scale ranging from 0 "not at all trustworthy" to 3 "a lot trustworthy". We adopt general trust measures that is used in related literature (e.g. [39, 40]) to our specific context by focusing on health-related information. Moreover, we constructed two personality traits indicators (agreeableness and openness to

new experiences) using 'Ten Item Personality Index (TIPI)' personality test items. The TIPI is an established test, a short version based on the five factor model of personality that has been validated in several countries [41]. Cognitive skills are measured in terms of numeracy using three items from an established module [42]. Respondents had to solve three mathematical tasks. Our measure of cognitive skills captures the share of correctly solved tasks. We measure the extent of social media usage in terms of the time respondents spent on social media in the last week and construct the following binary indicators for social media usage: a) less than 1 hour, b) between 1 and 10 hours, c) between 11 and 20 hours, and d) more than 20 hours.

Lastly, we also consider vaccine-related aspects: 'Vaccination status' refers to the share of vaccinations the respondent had received against the following three diseases: HPV, COVID-19, and polio. 'Vaccination knowledge' is a composite index that was derived by the responses to the following two statements: "Vaccination against tetanus has to be refreshed regularly to stay effective." and "The measles vaccines used in my country in the last 10 years used mRNA technology." 'Vaccine hesitancy' is an index that we constructed on the basis of the following three statements: (i) "I believe that governmental regulations in my country ensure quality vaccines and drugs.", (ii) "I believe that vaccines often cause more harm than good." and (iii) "I believe that Western countries use pharmaceutical companies to exploit African people for their own purposes.". The first item enters the index in reverse form. The items are adjusted versions of the vaccine readiness scale [43]. Respondents answered on a 7-point Likert scale ranging from 0 "strongly disagree" to 6 "strongly agree". Both indices were constructed using principal component analysis. The questionnaire including all survey items is presented in S1 Appendix.

## 2.5 Empirical strategy

Our analyses start with describing detection rates and statistics related to the intentional and actual sharing of misinformation in our sample.

Second, we run OLS regressions to estimate the determinants of deliberate and accidental fake news sharing actions on Facebook. More specifically, we estimate Eq 1 below:

$$Y_{icv} = \mathbf{X}`_{ic}\beta + \eta_v + \mu_c + \epsilon_{icv} \tag{1}$$

where $Y_{icv}$ is a binary indicator that identifies whether respondent $i$ in country $c$ has shared the presented article about vaccine $v$ accidentally or deliberately. $\mathbf{X}_{ic}$ is a matrix of individual-level characteristics including socio-demographics, personality traits, economic preferences, and vaccination-related attitudes, knowledge, and past behavior. The matrix also includes binary indicators related to an individual's assignment in another survey experiment. $\eta_v$ describes vaccine fixed effects, $\mu_c$ country fixed effects, and $\epsilon_{iv}$ is the error term. Our main specifications use heteroskedastic-robust standard errors. Since covariates are not randomly assigned we do not claim to estimate causal effects since we cannot completely rule out omitted variable bias. However, given that were are able to control for various factors we think that our results are able to reliably identify the determinants of accidental and deliberate sharing of fake news. All analyses were conducting in Stata 18.

## 3 Results

### 3.1 Detection of misinformation and sharing intention vs. action

In our sample, 54.7% of respondents correctly detected that the article contained misinformation. S3 Table in the supplementary information shows that in particular women, respondents

with better cognitive skills, and persons with lower levels of institutional trust were more likely to detect misinformation.

Examining sharing intentions, we find that about 52.1% of our respondents stated that they would be willing to share the article on their Facebook account (binary indicator). The intention to share question was directly asked after respondents had read the article. In Column 3 of S3 Table, we show that, ceteris paribus, in particular respondents who are older, male, richer, risk-loving and those with higher levels of institutional trust and lower cognitive skills were more willing to share the fake news article.

Turning towards our main outcome measure: About 13.8% of respondents shared the article on Facebook. Respondents could share the article on Facebook directly after they had declared their sharing intention. The results suggest that substantially less people actually share fake news articles compared to those who stated their intention to do so. Relative to the results of studies in the U.S. that observed actual sharing behavior, it should be noted that actual sharing rates in our study are substantially higher [2, 25, 27].

Before discussing our main results, we distinguish between accidental and deliberate distributors for both sharing indicators (intention vs. action). Regarding the intentional measure, about 40% of respondents can be classified as deliberate and about 60% as accidental distributors of misinformation. With respect to actual article sharing, we obtain very similar results, though the relative share tends to shift towards even higher rates of accidental distribution (63% vs. 37%). Overall, we can conclude that the majority of fake news distribution in our sample occurred accidentally, while a non-negligible number of respondents is purposefully spreading misinformation via social media.

## 3.2 Individual-level determinants of sharing patterns

In this section we investigate respondents' characteristics that are related to the accidental vs. deliberate forwarding of misinformation. Adopting a linear probability model and multivariate regression framework (OLS), we estimate what determines sharing intentions & actions. Our independent variables of interest include socio-demographic factors (age, gender, marital status, education, employment status, wealth, cognitive skills), personality traits (agreeableness and openness), economic preferences (trust and risk taking), social media usage, and vaccine-specific indicators (vaccination status, vaccine knowledge, vaccine attitudes). All regressions include country and vaccine-type (polio, HPV or COVID-19) fixed effects. We use robust standard errors.

**Actual sharing of fake news.** Column 1 of Table 1 presents the individual determinants of actual fake news sharing in general, while columns 2 and 3 contain our main results focusing on the determinants of deliberate and accidental sharing.

First, referring to column 1, our results show that the likelihood to share misinformation increases considerably with the age of the respondent. Relative to the reference group of persons age 18–29, respondents age 30–39 (50+) are about 7.4 (12.3) percentage points more likely to share the fake news article. Moreover, respondents who are male, in employment, more risk-loving, exhibit higher levels of trust in institutions, and have a history of past vaccinations are also more likely to share misinformation.

With respect to the determinants of deliberate sharing (column 2), we find that in particular older persons, respondents who are in employment and risk-loving individuals are more likely to share fake news deliberately. Next, we turn to the determinants of accidental fake news sharing. Column 3 reports that respondents that share misinformation accidentally are, on average, more likely to be older, male, and have higher levels of trust in institutions. Moreover, we

**Table 1. Determinants of deliberate and accidental fake news sharing.**

| | Sharing action | Deliberate sharing | Accidental sharing |
|---|---|---|---|
| | (1) | (2) | (3) |
| Age 30–39 | 0.074 *** | 0.032 *** | 0.041 *** |
| | (0.013) | (0.009) | (0.010) |
| Age 40–49 | 0.111 *** | 0.047 *** | 0.064 *** |
| | (0.022) | (0.015) | (0.018) |
| Age 50+ | 0.123 *** | 0.045 ** | 0.078 *** |
| | (0.031) | (0.020) | (0.026) |
| Female | -0.042 *** | -0.005 | -0.037 *** |
| | (0.009) | (0.006) | (0.008) |
| Married | -0.003 | -0.013 * | 0.010 |
| | (0.012) | (0.008) | (0.010) |
| No or primary education | -0.035 | 0.001 | -0.036 |
| | (0.030) | (0.021) | (0.022) |
| Secondary education | 0.025 ** | 0.008 | 0.017 * |
| | (0.011) | (0.007) | (0.009) |
| (Self-)employed | 0.036 *** | 0.019 *** | 0.017 ** |
| | (0.010) | (0.006) | (0.008) |
| Rich | 0.024 | -0.004 | 0.028 * |
| | (0.018) | (0.011) | (0.016) |
| Poor | 0.003 | 0.006 | -0.003 |
| | (0.012) | (0.008) | (0.010) |
| Cognitive skills | -0.000 | 0.004 | -0.004 |
| | (0.006) | (0.004) | (0.005) |
| Social media: < 1h last week | -0.017 | -0.016 | -0.001 |
| | (0.017) | (0.010) | (0.015) |
| Social media: 11–20h last week | -0.015 | -0.007 | -0.008 |
| | (0.012) | (0.008) | (0.010) |
| Social media: > 20h last week | -0.016 | -0.001 | -0.014 |
| | (0.012) | (0.008) | (0.010) |
| Agreeableness | 0.000 | -0.002 | 0.002 |
| | (0.002) | (0.001) | (0.002) |
| Openness | -0.004 | -0.003 * | -0.001 |
| | (0.002) | (0.002) | (0.002) |
| Risk taking | 0.009 ** | 0.006 *** | 0.003 |
| | (0.004) | (0.002) | (0.003) |
| Trust in institutions | 0.020 ** | -0.005 | 0.025 *** |
| | (0.008) | (0.005) | (0.007) |
| Vaccination | 0.012 * | 0.003 | 0.009 |
| | (0.007) | (0.005) | (0.006) |
| Vaccine knowledge | 0.004 | 0.001 | 0.003 |
| | (0.007) | (0.005) | (0.006) |
| Vaccine hesitancy | -0.006 | 0.001 | -0.007 ** |
| | (0.004) | (0.003) | (0.003) |
| Observations | 5,307 | 5,307 | 5,307 |

(*Continued*)

**Table 1.** (Continued)

| | Sharing action | Deliberate sharing | Accidental sharing |
|---|---|---|---|
| | (1) | (2) | (3) |
| $R^2$ | 0.043 | 0.020 | 0.032 |

Note: The table reports coefficient estimates and standard errors from OLS regressions. Regressions include fixed effects for vaccine-type, treatment assignment, and country. Standard errors are robust.

***,*,* denote significance at 1, 5 and 10%.

observe that persons with vaccine-hesitant attitudes are less likely to share fake news accidentally (about 0.7 percentage points).

As robustness checks, we estimate probit models instead of a linear probability models, presented in S4 Table, and provide results from OLS regressions in which standard errors are clustered at the country-vaccine-type level (S5 Table). To address concerns related to language barriers and misunderstanding, we run the analysis with a sub-sample that reports good to very good English language skills as a robustness check in S6 Table excluding N = 148 observations. While the results change somewhat, the overall findings are consistent across the different model specifications.

**Differences between intention vs. actual sharing measures.** In the following we explore differences in the determinants of intentional vs. actual sharing behaviour. Since in many contexts policy conclusions are derived from intentional measures only—given the absence of actual sharing data—the subsequent analysis highlights to what extent conclusions differ depending on the used measure. Table 2 depicts the respective results with columns 1–3 (4–6) showing results related to the deliberate (accidental) sharing of fake news.

Regarding the deliberate sharing of fake news articles columns 1 displays the determinants of the intention to share with column 3 showing the determinants of actual sharing behavior (conditional of having stated the intention to do so). Our main focus is on column 2 that shows the determinants of what we refer to as inconsistent behaviour (stating the intention to share the fake news article but ultimately not sharing it). We find that in particular men and more educated persons are more likely to exhibit inconsistent behavior regarding the deliberate sharing of the fake news article.

Turning towards the accidental sharing of fake news column 5 contains our results with respect to inconsistent behavior. We find that, ceteris paribus, especially younger individuals, men, persons with lower levels of education and cognitive skills, individuals with high levels of trust, and less vaccine hesitant persons are associated with inconsistent behavior with respect to the accidental sharing of the fake news article.

## 4 Discussion

This study investigates (i) the propensity to share vaccine-related fake news in SSA and (ii) the drivers behind the deliberate and accidental sharing of these articles. In online surveys with social media users in six SSA countries, we show study participants an article about a new vaccine (polio or HPV or COVID-19) that explicitly entails misinformation regarding its side-effects (infertility claim). Subsequently, we measure respondents' intention to share and actual sharing behaviour on Facebook. Likewise, we receive from each respondent an assessment of the accuracy of the article content.

The first part of our analysis derives three main findings. First, actual sharing rates of fake news articles are substantially higher in the context of health and SSA compared to estimates

**Table 2. Individual characteristics of deliberate and accidental fake news distributors.**

| | Deliberate intention | | | Accidental intention | | |
|---|---|---|---|---|---|---|
| | **(1)** | **(2)** | **(3)** | **(4)** | **(5)** | **(6)** |
| | intention | intention only | intention + sharing | intention | intention only | intention + sharing |
| Age 30–39 | 0.039 *** | 0.003 | 0.036 *** | -0.014 | -0.053 *** | 0.038 *** |
| | (0.015) | (0.013) | (0.008) | (0.016) | (0.014) | (0.010) |
| Age 40–49 | 0.061 ** | 0.007 | 0.054 *** | -0.005 | -0.065 *** | 0.060 *** |
| | (0.025) | (0.022) | (0.015) | (0.026) | (0.022) | (0.018) |
| Age 50+ | 0.039 | -0.007 | 0.046 ** | -0.028 | -0.102 *** | 0.074 *** |
| | (0.033) | (0.029) | (0.019) | (0.034) | (0.028) | (0.026) |
| Female | -0.034 *** | -0.027 ** | -0.007 | -0.095 *** | -0.059 *** | -0.035 *** |
| | (0.012) | (0.011) | (0.006) | (0.013) | (0.011) | (0.008) |
| Married | 0.000 | 0.016 | -0.016 ** | 0.053 *** | 0.043 *** | 0.010 |
| | (0.014) | (0.013) | (0.008) | (0.015) | (0.013) | (0.010) |
| No or primary education | -0.070 ** | -0.065 ** | -0.005 | 0.061 | 0.094 ** | -0.033 |
| | (0.035) | (0.032) | (0.019) | (0.046) | (0.044) | (0.022) |
| Secondary education | -0.020 | -0.026 ** | 0.006 | 0.061 *** | 0.043 *** | 0.017 * |
| | (0.013) | (0.012) | (0.007) | (0.015) | (0.014) | (0.009) |
| (Self-)employed | 0.009 | -0.007 | 0.016 ** | 0.015 | -0.002 | 0.017 ** |
| | (0.012) | (0.011) | (0.006) | (0.013) | (0.012) | (0.008) |
| Rich | -0.028 | -0.021 | -0.006 | 0.086 *** | 0.062 *** | 0.025 |
| | (0.021) | (0.019) | (0.010) | (0.024) | (0.023) | (0.016) |
| Poor | 0.013 | 0.012 | 0.001 | -0.000 | 0.002 | -0.002 |
| | (0.016) | (0.014) | (0.008) | (0.016) | (0.015) | (0.010) |
| Cognitive skills | -0.007 | -0.011 | 0.003 | -0.049 *** | -0.045 *** | -0.004 |
| | (0.007) | (0.007) | (0.004) | (0.008) | (0.007) | (0.005) |
| Social media: < 1h last week | -0.022 | -0.009 | -0.013 | 0.038 | 0.039 * | -0.001 |
| | (0.021) | (0.019) | (0.010) | (0.024) | (0.022) | (0.015) |
| Social media: 11–20h last week | 0.014 | 0.018 | -0.004 | -0.014 | -0.005 | -0.010 |
| | (0.015) | (0.014) | (0.007) | (0.015) | (0.014) | (0.010) |
| Social media: > 20h last week | -0.021 | -0.021 | 0.001 | -0.005 | 0.010 | -0.015 |
| | (0.015) | (0.013) | (0.008) | (0.016) | (0.015) | (0.010) |
| Agreeableness | 0.001 | 0.002 | -0.001 | -0.002 | -0.004 | 0.002 |
| | (0.003) | (0.002) | (0.001) | (0.003) | (0.003) | (0.002) |
| Openness | 0.001 | 0.002 | -0.002 | -0.008 ** | -0.006 ** | -0.001 |
| | (0.003) | (0.003) | (0.002) | (0.003) | (0.003) | (0.002) |
| Risk taking | 0.007 * | 0.001 | 0.006 *** | 0.009 * | 0.006 | 0.003 |
| | (0.004) | (0.004) | (0.002) | (0.005) | (0.005) | (0.003) |
| Trust in institutions | -0.011 | -0.008 | -0.004 | 0.074 *** | 0.049 *** | 0.025 *** |
| | (0.010) | (0.009) | (0.005) | (0.011) | (0.010) | (0.007) |
| Vaccination | 0.007 | 0.002 | 0.005 | 0.025 *** | 0.018 ** | 0.008 |
| | (0.009) | (0.008) | (0.004) | (0.009) | (0.009) | (0.006) |
| Vaccine knowledge | 0.010 | 0.006 | 0.004 | -0.005 | -0.008 | 0.004 |
| | (0.009) | (0.008) | (0.005) | (0.010) | (0.009) | (0.006) |
| Vaccine hesitancy | 0.004 | 0.003 | 0.000 | -0.033 *** | -0.025 *** | -0.008 ** |
| | (0.005) | (0.004) | (0.003) | (0.005) | (0.005) | (0.003) |
| Observations | 5,307 | 5,307 | 5,307 | 5,307 | 5,307 | 5,307 |

(*Continued*)

**Table 2.** (Continued)

| | Deliberate intention | | | Accidental intention | | |
|---|---|---|---|---|---|---|
| | (1) | (2) | (3) | (4) | (5) | (6) |
| | intention | intention only | intention + sharing | intention | intention only | intention + sharing |
| $R^2$ | 0.017 | 0.011 | 0.022 | 0.073 | 0.055 | 0.031 |

Note: The table reports coefficient estimates and standard errors from OLS regressions. The regressions include fixed effects for vaccine-type, treatment assignment, and country. Standard errors are robust.

\*\*\*,\*,\* denote significance at 1, 5 and 10%.

reported in studies on political topics in rich countries. For instance, Guess et al. [27] report that about 8.5% of respondents in the U.S. shared at least once misinformation on Facebook during the presidential election campaign in 2016, while our study documents a sharing rate of 13.8%. Second, we demonstrate that the majority (around two thirds) of fake news articles is disseminated accidentally which is in line with studies from richer countries [23]. Third, we document massive differences across the adopted measures of fake news sharing (intention to share vs. actual sharing) with actual sharing rates being much lower compared to those derived from intention-based measures.

The second part of our analysis identifies the determinants of deliberate and accidental sharing of fake news articles on Facebook. With respect to the actual sharing of the fake news article our results suggest that being older, risk-loving, and in employment increases the likelihood to deliberately share fake news. Furthermore, being older, male, in employment, and exhibiting high levels of trust in institutions is positively associated with sharing fake news accidentally.

Overall, our results indicate that older, male, risk-loving, trusting persons, and those in employment are more likely to distribute vaccine-related misinformation. In particular, our study confirms the importance of age in spreading misinformation. The results show that while older persons are more likely to detect misinformation, they are still more likely to share misinformation both accidentally and deliberately. A further interesting insight concerns the relationship between trust and accidental sharing, hinting towards the use of the information source as a credibility signal in the decision to share the article. Such heuristics help to rapidly take decisions, however, may trick persons into sharing misinformation particular when misinformation is provided by seemingly trustworthy sources.

Assessing the potential measurement error of intentional measures, our study documents biases in the individual determinants of deliberate and accidental sharing. The documented differences between the two sharing measures (intentional and actual sharing behavior) challenge the validity of studies using self-reported behavioral intentions for making claims about actual sharing behaviour.

We believe that this study can help policymakers to more efficiently target relevant groups to limit the spread of misinformation. In particular, our results imply that older social media users with employment were relatively more prone to circulate health-related misinformation in our context. Additionally, we find a strong relationship between trust in institutions and the accidental distribution of misinformation, highlighting that official sources should particularly pay attention to distributing valid and truthful information.

Lastly, we would like to point out some limitations of our study. First, respondents predominantly stem from one country, Kenya, which potentially reduces the external validity of our results for other LMICs. Consequently, empirical evidence from other regions is desirable in

order to better understand the generalizibility of our findings. Second, we cannot fully rule out that the action of sharing itself has biased respondents' perception of the accuracy of the article content. Though the survey item on the perceived reliability of the article comes at a much later stage in the online survey, respondents still might remember their behavior and adjust their perception or response accordingly (consistency bias) [22]. If such a bias were to exist in our study context it would, however, not affect any of the results related to the determinants of the overall sharing action of the fake news article. Third and bearing in mind that covariates are not randomly distributed, the study does not provide causal evidence on the impact of a certain variable on deliberate and accidental fake news sharing actions. Nonetheless, the study adopts a well-established framework to elicit the determinants of the related fake news sharing actions.

## 5 Conclusion

Limitations notwithstanding, our study shows that the sharing of vaccine misinformation is prevalent in SSA countries, yet to a lesser extent as suggested by standard survey measures on sharing intentions. While the majority of health misinformation is shared accidentally, a not negligible part purposefully distributes misinformation. Analysing the individual determinants of fake news distributors, our study reveals that older, male, risk-loving, trusting persons have a higher likelihood to share misinformation in our context. Overall, our results shed light on the detection and sharing of health misinformation in a realistic online setting, providing novel insights on who is susceptible to fall for and more likely to disseminate fake news.

## Supporting information

**S1 Table. Summary statistics.** This file provides the summary statistics of the study sample.
(PDF)

**S2 Table. Observations by country.** This file provides the descriptive distribution of respondents per country.
(PDF)

**S3 Table. Detection of misinformation.** This file provides the regression results of individual characteristics associated with detecting misinformation.
(PDF)

**S4 Table. Alternative regression model: Probit.** This file provides the regression results of the main analysis using a probit instead of an OLS regression estimation.
(PDF)

**S5 Table. Alternative regression model: Clustered standard errors.** This file provides the regression results of the main analysis using alternative specifications of standard errors.
(PDF)

**S6 Table. Robustness check: English language skills.** This file provides the regression results of the main analysis with a sub-sample that reports very good or good English language skills.
(PDF)

**S1 Appendix. Survey questionnaire: This file contains the survey questionnaire.**
(PDF)

**S2 Appendix. Fake news articles: This file contains the articles used in the experiment.**
(PDF)

**S1 Data.**

(ZIP)

## Author Contributions

**Conceptualization:** Kerstin Unfried, Jan Priebe.

**Formal analysis:** Kerstin Unfried.

**Investigation:** Kerstin Unfried.

**Methodology:** Kerstin Unfried.

**Project administration:** Kerstin Unfried, Jan Priebe.

**Supervision:** Jan Priebe.

**Visualization:** Kerstin Unfried.

**Writing – original draft:** Kerstin Unfried.

**Writing – review & editing:** Jan Priebe.

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
