## [Decision Letter · Decision Letter 0]

18 Dec 2023

PONE-D-23-32351Who shares fake news on social media? Evidence from vaccines and infertility claims in sub-Saharan AfricaPLOS ONE

Dear Dr. Unfried,

Thank you for submitting your manuscript to PLOS ONE. After careful consideration, we feel that it has merit but does not fully meet PLOS ONE’s publication criteria as it currently stands. Therefore, we invite you to submit a revised version of the manuscript that addresses the points raised during the review process.

We look forward to receiving your revised manuscript.

Kind regards,

Amir H. Pakpour, Ph.D.

Academic Editor

PLOS ONE

Journal Requirements:

Reviewers' comments:

Reviewer's Responses to Questions

**Comments to the Author**

1. Is the manuscript technically sound, and do the data support the conclusions?

Reviewer #1: Partly

2. Has the statistical analysis been performed appropriately and rigorously? 

Reviewer #1: Yes

3. Have the authors made all data underlying the findings in their manuscript fully available?

Reviewer #1: Yes

4. Is the manuscript presented in an intelligible fashion and written in standard English?

Reviewer #1: Yes

5. Review Comments to the Author

Reviewer #1: Although I appreciate the authors' attempt and hard efforts on this topic, which shed light regarding the actual behaviors in sharing fake news, I think that the present study has some issues. Please see my comments below.

1. Some papers on the rumors associated with COVID-19 and social trust in government/health authorities should be used in the Introduction to illustrate the importance of tackling fake news. Please see the following papers.

Kar B, Kar N, Panda MC. Social trust and COVID-appropriate behavior: Learning from the pandemic. Asian J Soc Health Behav 2023;6:93-104

Nurmansyah MI, Suraya I, Fauzi R, Al-Aufa B. Beliefs about the effects of smoking on corona virus disease 2019 and its impact on the intention to quit and smoking frequencies among university students smokers in Jakarta, Indonesia. Asian J Soc Health Behav 2023;6:7-13

2. A big issue for me is that the authors let the participants shared the fake news. I think that this is unethical although this is an important measure. Therefore, I would like to know if the authors have do anything to stop sharing the fake news to avoid negative consequence.

3. The measures used in the present study did not have psychometric evidence. I wonder how the authors developed these measures and how they ensure their reliability and validity for use.

4. I suppose that the authors used English for the study measures. How did the authors ensure that the participants had good English to complete the questions? For example, I see that the authors collected participants receiving no or primary education; how do the authors ensure that these participants could understand the questionnaires?

5. In my opinion, Figures 1 and 2 are redundant as their information can be briefly mentioned in the Results section.

6. PLOS authors have the option to publish the peer review history of their article (what does this mean?). If published, this will include your full peer review and any attached files.

Reviewer #1: No

---

## [Author Response · Author response to Decision Letter 0]

31 Jan 2024

Please see the comments to the reviewers in the response to reviewer document. We have carefully addressed all the comments.

---

## [Decision Letter · Decision Letter 1]

24 Mar 2024

Who shares fake news on social media? Evidence from vaccines and infertility claims in sub-Saharan Africa

PONE-D-23-32351R1

Dear Dr. Unfried,

We’re pleased to inform you that your manuscript has been judged scientifically suitable for publication and will be formally accepted for publication once it meets all outstanding technical requirements.

Kind regards,

Amir H. Pakpour, Ph.D.

Academic Editor

PLOS ONE

Additional Editor Comments (optional):

Reviewers' comments:

Reviewer's Responses to Questions

**Comments to the Author**

1. If the authors have adequately addressed your comments raised in a previous round of review and you feel that this manuscript is now acceptable for publication, you may indicate that here to bypass the “Comments to the Author” section, enter your conflict of interest statement in the “Confidential to Editor” section, and submit your "Accept" recommendation.

Reviewer #1: All comments have been addressed

2. Is the manuscript technically sound, and do the data support the conclusions?

Reviewer #1: Yes

3. Has the statistical analysis been performed appropriately and rigorously? 

Reviewer #1: Yes

4. Have the authors made all data underlying the findings in their manuscript fully available?

Reviewer #1: Yes

5. Is the manuscript presented in an intelligible fashion and written in standard English?

Reviewer #1: Yes

6. Review Comments to the Author

Reviewer #1: The authors have satisfactorily addressed all my prior concerns. I am happy with the present revision and do not have any more comments.

7. PLOS authors have the option to publish the peer review history of their article (what does this mean?). If published, this will include your full peer review and any attached files.

Reviewer #1: No

---

## [Editor Report · Acceptance letter]

28 Mar 2024

PONE-D-23-32351R1 

PLOS ONE

Dear Dr. Unfried, 

I'm pleased to inform you that your manuscript has been deemed suitable for publication in PLOS ONE. Congratulations! Your manuscript is now being handed over to our production team.

Kind regards, 

on behalf of

Dr. Amir H. Pakpour 

Academic Editor

PLOS ONE